# A Clinical Trial of the Effect of a Blood Leakage Detection Device for Patients during Hemodialysis

**DOI:** 10.3390/ijerph16132388

**Published:** 2019-07-05

**Authors:** Yang-Kun Ou, Ming-Jui Wu, Wei-Siang Ciou, Yi-Chun Du

**Affiliations:** 1Department of Creative Product Design, Southern Taiwan University of Science and Technology, No. 1, Nan-Tai Street, Yungkang District, Tainan 71005, Taiwan; 2Department of Internal Medicine, Kaohsiung Veterans General Hospital Tainan Branch, No. 427, Fuxing Road, Yongkang District, Tainan 71051, Taiwan; 3Department of Electrical Engineering, Southern Taiwan University of Science and Technology, No. 1, Nan-Tai Street, Yungkang District, Tainan 71005, Taiwan

**Keywords:** blood leakage detection device, hemodialysis (HD), quality of care, risk control

## Abstract

In hemodialysis, vascular access is usually achieved through an arteriovenous fistula, and a dislodged needle can cause varying degrees of injury to patients. In severe cases, the loss of blood can prove to be fatal. This study proposed a blood leakage detection device for patients during hemodialysis (HD). First, the device was tested on a phantom arm, and later in a clinical test on patients receiving HD. The thoughts of the patients and the nursing staff involved were surveyed before and after the introduction of the device. Analysis of the results indicated that the device achieved 100% and 98.9% accuracy rates on the phantom arm test and clinical test, respectively. The results suggested that patients believed the device could reduce their mental anxiety, and the nursing staff considered the device reliable and that it would enhance the quality of care. The proposed detection device can be extended to similar applications for preventing catheter dislodgement, and to improve patient safety and reduce the stress of clinical nursing staff.

## 1. Introduction

Statistics from Taiwan’s National Health Insurance Administration show that as of 2016, the incidence and prevalence rates of kidney disease in Taiwan had become the highest in the world, with 78,000 patients on hemodialysis (HD) or peritoneal dialysis, indicating a prevalence rate of 3177.8 per million people [1]. Although the incidence rate of kidney disease has been gradually decreasing in Taiwan, the prevalence rate of HD has risen steadily, owing to the following reasons: (1) population aging and an increase in patients with chronic diseases such as hypertension, hyperlipidemia, and hyperglycemia; (2) provision of full insurance coverage for dialysis by the National Health Insurance and lowering of the coverage threshold for such treatments; (3) excellent dialysis treatment quality; and (4) a low kidney transplant rate [2].

Patients on HD must receive treatment 3 times per week and for 4 hours per treatment session, in order to maintain the basic kidney function necessary for survival [3]. HD is a common alternative treatment in Taiwan, and medical personnel are well-practiced at intravenous insertion; however, venous needle dislodgement (VND) still occurs occasionally, which can lead to blood leakage. VND is most commonly caused by improper arm movement in the patients, and is particularly common among agitated patients. When this occurs with patients of advanced age or in a state of disturbed consciousness, these patients may barely notice that the needle has been dislodged and they have blood leakage. Therefore, for such patients, constant inspection by medical personnel and caregivers is indispensable. Furthermore, although dialysis machines usually have a built-in mechanism for detecting blood leakage, the presence of sediment in the system can sometimes cause false alarms. Moreover, when blood leakage occurs from a VND, the machines usually require 3−5 minutes to detect it, and sometimes they even fail to detect it at all because the puncture is small and the leakage is slow. As a consequence, the patients may sustain varying degrees of injury. For this reason, care should be taken during dialysis to prevent the needle from being covered by blankets or clothing, to make the detection of VND easier for medical personnel, caregivers, and the patients themselves.

The 2016 annual report of the Taiwan Patient Safety Reporting System listed the dislodgement of needles, catheters, and similar devices as third among the 13 most common incident types that threaten patient safety (the top two being drug- and fall-related incidents). In addition, it was reported that HD-related device dislodgement alone accounted for 357 incidents in 2016 [4]. A survey by Axley et al. [5] reported that up to 76.6% of respondents had witnessed VND during HD in the previous 5 years, 57.9% indicated that they were concerned about VND very often or often, and 85.3% considered education material beneficial for the reduction of VND risk. Two studies published in 2005 warned that although VND is not an extremely common occurrence, it could be life-threatening or even cause death [6,7]. In 2012, the American Nephrology Nurses Association established a team to examine the occurrence and consequences of VND, as well as to propose measures for medical personnel, patients with kidney disease, and patients’ families to prevent VND. All of these studies suggested that although the likelihood of VND directly causing death is extremely small, it can, nevertheless, have fatal consequences. This is particularly true for patients who are in an agitated state, have impaired cognitive abilities, or are receiving HD alone.

Numerous factors impose massive pressure on nursing staff and affect their retention rate and personal health; these include a sense of responsibility to medical services and to life, as well as clinical situations such as workload, nurse–patient relationships, irregular work schedules, role-specific conflicts and requirements, administrative work, personnel turnover, work environment, interpersonal relationships, and lack of professional knowledge and skills [8,9,10,11,12,13]. Due to the unique roles and importance of clinical nursing staff, excessive pressure not only affects their personal health, but also the quality of the clinical services they provide. Hence, medical institutions worldwide strive to improve working conditions for nursing staff and mitigate their stress without sacrificing nursing quality.

Thus, in order to provide an additional layer of protection for patients undergoing HD, as well as to alleviate the clinical pressure of nursing staff and facilitate positive nurse–patient interactions, a blood leakage detection device has been developed. The present study describes device testing in both a simulated and clinical setting, and users’ responses from nursing staff and patients.

## 2. Materials and Methods

The design of the blood leakage detection device was based on the research team’s discussions with physicians and nurses working in the HD unit. Details of the device components and operation are described by Du et al. [14]. Testing of the device was first simulated on a phantom arm and, subsequently, on human volunteers. The methods applied were as follows.

### 2.1. Phantom Arm Simulation Experiment 

Phantom arms are one of the most commonly used teaching aids in basic clinical practices. For this experiment, a simulated blood circulation system was adopted for the phantom arm to simulate blood flow in a real arm during HD (Figure 1). This system consisted of a single-phase 110 V/60 Hz adjustable-speed motor (100−500 mL/min) (Orientalmotor, Speed Control Motor, New Taipei, Japan), and an AC110/60 Hz adjustable timer (ANLY, TRD-NC relay, Taipei, Taiwan) relay with a 0.2 s reset time (adjustable range of 0.2–5 s), enabling the system to simulate pulses of 50−180 beats per minute. In order to verify the feasibility and accuracy of the simulation system, a flow meter and a Doppler ultrasound were also connected to it to record and monitor flowrate and heart rate variability. Du et al. [14] performed tests on the system’s functions and found that the simulated blood flow velocity was similar to the default value of the dialysis machine, therefore verifying that the system could effectively simulate the conditions of HD.

Besides blood flow velocity, the sensitivity of the detection system was also tested. After the arm was punctured, a syringe was used to eject blood to simulate blood leakage. When the device detects a leakage of blood, an alarm goes off to notify the care staff to attend to the patient. In order to test for the sensitivity of the blood leakage detection system, blood was set to leak at amounts of 0.1, 0.2, 0.3, and 0.5 mL (Figure 2) at angles of 0°, 36°, 72°, 108°, 144°, 180°, 216°, 252°, 288°, and 324° in relation to the needle—hereafter referred to as angles 1–10, respectively (Figure 3), and the alarm was triggered when blood leakage was detected at any angle. 

### 2.2. Clinical Test on Human Volunteers

#### 2.2.1. Participants

Our study was approved by the Research Ethics Committee of Kaohsiung Veterans General Hospital, Tainan Branch, and informed consent was obtained from each study participant. A total of 11 (9 male and 2 female) HD patient volunteers participated in the clinical test. All of the HD participants had an existing schedule of 3 sessions of HD a week, and already had an implanted arteriovenous fistula (AVF) for vascular access before the start of the clinical test. The procedure and purpose of the experiment were explained to the volunteers before the 5 month experiment commenced, and all volunteers signed a letter consenting to their participation. During the clinical testing period, the detection device was affixed to the skin directly on top of the AVF of the participants during HD session. Three items were recorded at each HD session: (1) whether the alarm of the detection device went off during the HD session, (2) nurses’ notes on whether blood leakage occurred during examination of the AVF at the end of the HD session, and (3) any adverse events related to the use of the detection device. To gather information about their subjective experience, HD participants were asked to fill in a questionnaire about the device. Seven members of the nursing staff were also recruited to fill in a questionnaire about the device during the clinical test.

#### 2.2.2. Research Instruments and Evaluation Methods

Questionnaire

Two sets of questionnaires that adopted a five-point Likert Scale were used to collect the thoughts of the HD patients and nursing staff on HD before and after the introduction of the detection device. The questionnaire for the HD patients comprised 13 items that covered the patients’ anxiety, quality of rest, and satisfaction. The questionnaire for the nursing staff comprised of 10 items that covered their thoughts on quality of care, complexity of device deployment, and device reliability. The items were designed to reflect the respondents’ subjective thoughts on the blood leakage detection device (Table 1). The questionnaires were filled in twice; once before introducing the detection device, and again after the experiment.

Evaluation methods

The classification output quality of the blood leakage detection device was evaluated using the *F measure* and accuracy rate. The *F measure* is the harmonic means of precision and recall, and is derived from the formula
$$F\;measure=\frac{2RP}{R+P},$$
where *R* represents *Recall*, the percentage of actual positive items that was classified out of all actual positive items (*Recall* is also known as sensitivity), and *P* represents *Precision*, the percentage of actual positive items out of all classified items (*Precision* is also known as positive predictive value). The formulae for *Recall* and *Precision* are:
$$Recall=\frac{TP}{TP+FP}$$
$$Precision=\frac{TP}{TP+FN}$$
where *TP* represents the number of true positives (accurately classified positive items), *FP* represents the number of false positives (inaccurately classified positive items), and *FN* represents the number of false negatives (relevant items inaccurately classified as negative). 

Therefore, the higher the *R* and *P* values, the higher the *F measure* value, indicating better classification output quality of the detection device.

The accuracy rate is the percentage of all blood leakage detection device classifications that were correct. It is derived from the following formula:
$$\mathrm{Accuracy}\;\mathrm{rate}=\frac{TP+TN}{TP+TN+FP+FN}$$

## 3. Results

### 3.1. Phantom Arm Simulation Experiment

In practice, a small amount of blood can leak out regardless of the blood flow velocity or extent of needle dislodgement, and this is particularly true with high blood flow velocity. According to the results of the phantom arm simulation test (Table 2), the detection device was able to detect blood leakage when the blood flow velocity was at 200 mL/min or above.

According to the results of the sensitivity test (Table 3), the device correctly detected blood leakage and did not send out false alarms regardless of the amount or angle of leakage. This shows that the detection performance of the proposed hollow and circular device is extremely satisfactory.

### 3.2. Clinical Test

Sex, age, and HD history of the participants are shown in Table 4. In this study, a total of 11 subjects with an average age and standard deviation (SD) of 65.0 ± 14.2 were recruited from the Kaohsiung Veterans General Hospital (KVGH), Tainan Branch. The clinical test design was approved by the KVGH Institutional Review Board (IRB No. VGHKS16-CT8-20). Although the clinical testing period was 5 months long and all subjects were scheduled for 3 HD sessions per week, some subjects joined the study later and, therefore, not everyone was tested for the full 5 months. Overall, the experiment accumulated 544 test data entries. No adverse reactions specific to the detection device such as increased patient stress, discomfort, or allergic reaction to the adhesive of the device occurred. During the test, a total of 73 blood leakage incidents occurred, of which 67 were detected by the device (Table 5). The device’s performance is further evaluated with indices of precision, recall, *F measure*, and accuracy rate, as shown in Table 6. The results indicated that the proposed blood leakage detection device showed favorable performance, and was proven to be capable of accurately detecting blood leakage.

Concerning the thoughts of the device’s users, results of a Wilcoxon signed-rank test revealed that, in terms of patients’ subjective thoughts, the device was able to effectively alleviate anxiety but did not exhibit any significant effect on quality of rest or satisfaction (Table 7). As for the nursing staff, the device appeared to be able to effectively improve the quality of care they provided and was regarded as reliable (Table 8).

## 4. Discussion

This study evaluated a blood leakage detection device in clinical tests and found its accuracy rate to be 98.9%. The device failed to detect blood leakage 6 times; the reason for this was that a tiny gap existed between the tape and the puncture, and hence, when an extremely small amount of leakage occurred, the blood was quickly absorbed by the gauze before it was able to trigger the alarm.

HD is a prolonged undertaking that requires patients to undergo treatment on a regular basis, during which they are subjected to extensive physiological, psychological, and social stress [15]. Moreover, the discomfort involved in the process can often cause patients to develop an aversion to the treatment [16]. Because of the importance of patient’s mood in managing disease and maintaining well-being, the design of the device intends to improve the HD experience for the patient. The device was found to alleviate the mental stress of patients on HD, allowing them to undergo the treatment in a relatively calm, relaxed, and stable state. Such a quality is considered beneficial to both patients’ conditions and the effectiveness of their treatment.

Nursing staff are subjected to high emotional labor and stress, which not only affects their physical and mental health, but also reduces their morale and work efficiency. When patients receive HD, nursing staff are required to check the patients on a regular basis to prevent VND. Taiwan has seen numerous cases of nurses being convicted because of VND [17], and such precedents are a form of work pressure for nurses. In severe cases, the pressure can even affect their work performance and lead to errors. The proposed blood leakage device was found to be reliable and capable of facilitating a positive nurse–patient relationship. Therefore, from a medical quality perspective, the device is beneficial for patient safety, nurse–patient relationships, and quality of care. 

In Taiwan, most HD patients undergo hemodialysis during the daytime, and many fall asleep during the 3–4 hour long session, during which the occasional motions and movements of the body or limbs is a great risk for VND. Furthermore, patients at high risk for being agitated or restless—such as elderly patients with dementia—require constant attention. There has even been a case of fatal hemorrhaging from VND where a long-term HD patient suffering from major depression pulled out his own needle. Reports such as these put considerable stress on HD clinical care personnel in Taiwan. Although the HD staff-to-patient ratio is 1:4, the nursing staff may need to temporarily leave for other clinical duties and not be able to remain at the patient’s side throughout the entire HD session. Therefore, this device can provide significant help to improve patient safety.

One major purpose of product design was to provide patients undergoing HD with further safety and convenience in their lives. In the medical industry, the development of assistive medical devices aims to improve the safety and convenience of medical services for both patients and medical personnel. The device proposed in this study uses flexible array sensing technology, and is made up of a green halogen-free and lead-free printed circuit board (PCB) and medical-grade plastic approved by the US Food and Drug Administration. Therefore, it not only conforms to standing medical regulations and EU standards, but also has excellent potential for extended clinical applications. Specifically, in addition to HD, use of this device may be expanded for the safety of patients to prevent the dislodgement of peripheral venous catheters.

## Figures and Tables

**Figure 1 ijerph-16-02388-f001:**
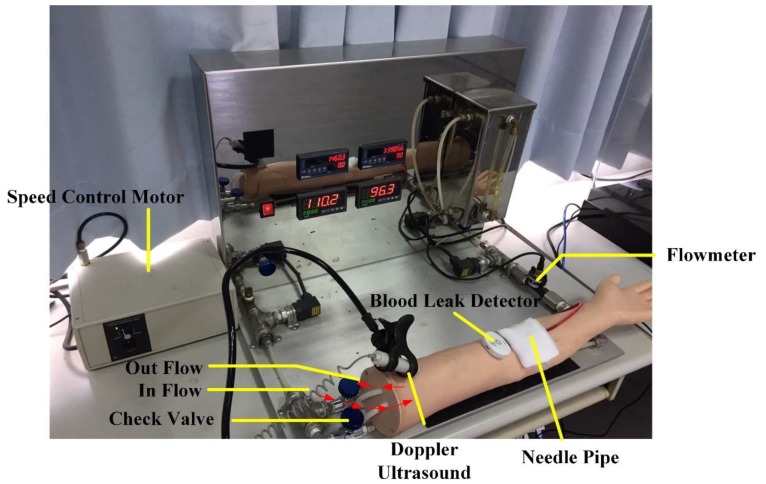
Venous needle dislodgement (VND) blood leakage test on a phantom arm.

**Figure 2 ijerph-16-02388-f002:**
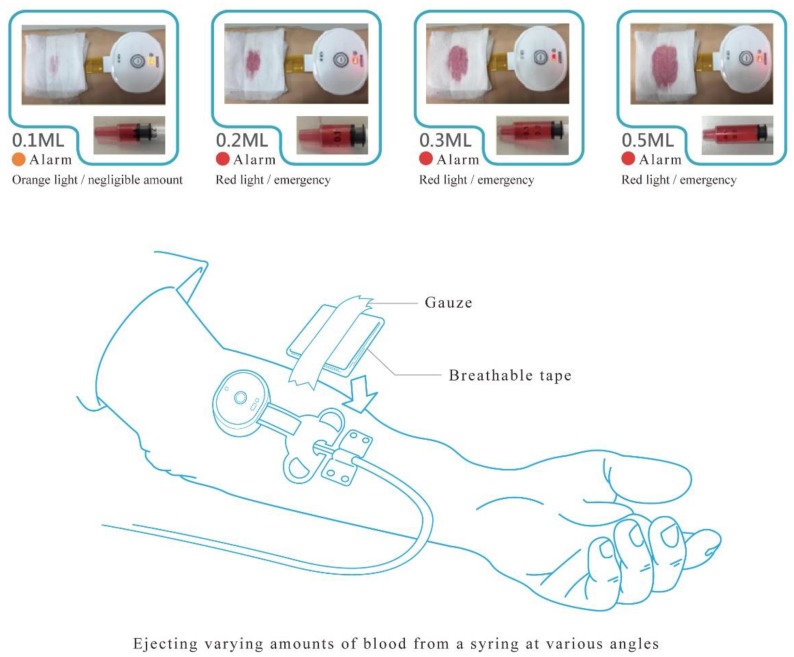
Blood leakage test.

**Figure 3 ijerph-16-02388-f003:**
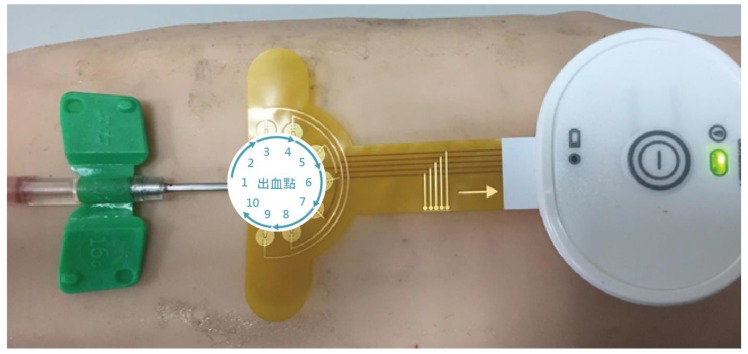
Angles of blood ejection.

**Table 1 ijerph-16-02388-t001:** Questionnaires for the patients and nursing staff.

Patients	Nursing Staff
I feel safe now.	I am able to provide the patients receiving HD with comprehensive care.
I feel nervous now.	I am able to pay attentive care to patients receiving HD.
I feel relaxed now.	I am satisfied with my interactions with patients receiving HD.
I am looking forward to the introduction of the blood leakage detection device.	The blood leakage detection device can affect my interactions with patients.
The detection device makes me feel safe.	The blood leakage detection device can increase my workload.
I am worried about blood leakage during HD right now.	The blood leakage detection device is complicated to install.
I am worried about other possible accidents during HD right now.	The blood leakage detection device is reliable.
HD is uncomfortable.	The blood leakage detection device makes me feel safe.
I feel fully rested during HD.	The blood leakage detection device can mitigate my stress when caring for patients.
I am satisfied with the HD.	I am looking forward to the introduction of the blood leakage detection device.
I am satisfied with the care I receive during HD.	
I am satisfied with the attentiveness of the nursing staff during HD.	
I am satisfied with the nurses’ interactions with me during HD.	

**Table 2 ijerph-16-02388-t002:** Results of the phantom arm test.

Blood Flow Velocity (mL/min)	Dislodged Needle Length (mm)	Blood Leakage (Yes/No)	Detection (Yes/No)
200	18	Y	Y
250	18	Y	Y
300	16	Y	Y
350	15	Y	Y
400	14	Y	Y

**Table 3 ijerph-16-02388-t003:** Results of the sensitivity test.

Simulated Amount of Blood Leakage	Angle	Alarm S/F	Simulated Amount of Blood Leakage	Angle	Alarm S/F
0.5 mL	1	10/0	0.2 mL	1	10/0
	2	10/0		2	10/0
	3	10/0		3	10/0
	4	10/0		4	10/0
	5	10/0		5	10/0
	6	10/0		6	10/0
	7	10/0		7	10/0
	8	10/0		8	10/0
	9	10/0		9	10/0
	10	10/0		10	10/0
0.3 mL	1	10/0	0.1 mL	1	10/0
	2	10/0		2	10/0
	3	10/0		3	10/0
	4	10/0		4	10/0
	5	10/0		5	10/0
	6	10/0		6	10/0
	7	10/0		7	10/0
	8	10/0		8	10/0
	9	10/0		9	10/0
	10	10/0		10	10/0

* S/F mean Success/Failure.

**Table 4 ijerph-16-02388-t004:** Demographic information of the participants.

**Case No.**	**Case 1**	**Case 2**	**Case 3**	**Case 4**	**Case 5**	**Case 6**	**Case 7**	**Case 8**
Sex	Female	Male	Male	Male	Female	Male	Male	Male
Age	47	69	85	84	65	49	57	88
HD history (years)	22	1.5	1.0	7.0	21	1.0	8.5	2.0
**Case No.**	**Case 9**	**Case 10**	**Case 11**					
Sex	Male	Male	Male					
Age	64	55	52					
HD history (years)	9.0	1.5	3.5					

**Table 5 ijerph-16-02388-t005:** Cluster of the classification output by the blood leakage detection device.

		True Condition	
		Condition Positive	Condition Negative
Predicted condition	Predicted condition positive	471	6
	Predicted condition negative	0	67

**Table 6 ijerph-16-02388-t006:** Performance of distinguishing blood leakage.

Precision	Recall	*F Measure*	Accuracy Rate
98.7%	100%	99.4%	98.9%

**Table 7 ijerph-16-02388-t007:** Patients’ subjective thoughts before and after using the device.

	Before	After	*p* Value
I feel safe now.	2.08	2.42	0.046 *
I feel nervous now.	2.42	2.08	0.046 *
I feel relaxed now.	2.08	2.42	0.046 *
I am looking forward to the introduction of the blood leakage detection device.	2.33	2.83	0.034 *
The detection device makes me feel safe.	2.42	2.92	0.014 *
I am worried about blood leakage during HD right now.	2.50	2.42	0.705
I am worried about other possible accidents during HD right now.	2.50	2.42	0.705
HD is uncomfortable.	2.42	2.58	0.414
I feel fully rested during HD.	2.33	2.67	0.102
I am satisfied with the HD.	2.67	2.83	0.317
I am satisfied with the care I receive during HD.	2.75	2.83	0.655
I am satisfied with the attentiveness of the nursing staff during HD.	2.75	2.83	0.655
I am satisfied with the nurses’ interactions with me during HD.	2.75	2.83	0.655

* *p* < 0.05

**Table 8 ijerph-16-02388-t008:** Nursing staffs’ subjective thoughts before and after using the device.

	Before	After	*p* Value
I am able to provide the patients receiving HD with comprehensive care.	1.43	3.00	0.015 *
I am able to pay attentive care to patients receiving HD.	1.29	3.00	0.014 *
I am satisfied with my interactions with patients receiving HD.	1.29	3.00	0.014 *
The blood leakage detection device can affect my interactions with patients.	2.00	2.29	0.746
The blood leakage detection device can increase my workload.	2.00	2.14	0.888
The blood leakage detection device is complicated to install.	2.14	2.00	0.739
The blood leakage detection device is reliable.	1.29	3.00	0.014 *
The blood leakage detection device makes me feel safe.	1.57	3.00	0.015 *
The blood leakage detection device can mitigate my stress when caring for patients.	1.71	2.86	0.038 *
I am looking forward to the introduction of the blood leakage detection device.	1.29	3.00	0.014 *

* *p* < 0.05

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
