# Peer review of "A Clinical Trial of the Effect of a Blood Leakage Detection Device for Patients during Hemodialysis"

_ijerph, 2019, doi:10.3390/ijerph16132388_

Round 1
Reviewer 1 Report
As a reviewer outside of your field, I am not sure what typical blood flow looks like/what is most commonly seen in dialysis. It would help strengthen your discussion/results if you compared your device to other existing devices and explained how your device fits into the current technology landscape. You have done this to some extent but have not used specific numbers to compare your device performance to others. Also you titled this paper "Design and Clinical Test..." but it appears you described the design of the device elsewhere and this was just the testing of the device. You may want to adjust the title so the purpose of the paper is more clear.
Author Response
Responses to Reviewer 1#
Paper Number | : | ijerph-538775 |
Paper Title | : | A Clinical Trial of the Effect of a Blood Leakage Detection Device for Patients during Hemodialysis |
Authors | : | Yang-Kun Ou, Ming-Jui Wu, Wei-Siang Ciou and Yi-Chun Du |
Corresponding author | : | Yi-Chun Du
|
Dear Supervisor,
We highly appreciate the reviewers’ valuable comments. The manuscript has been carefully revised accordingly. The manuscript has been revised and reformulated with good scientific rules of English to improve its quality and has been confirmed by a native English speaker. Notably, the revised and added contexts of the manuscript are marked in Red. The point-to-point responses are shown below.
For Reviewer 1#
1. Question 1: As a reviewer outside of your field, I am not sure what typical blood flow looks like/what is most commonly seen in dialysis. It would help strengthen your discussion/results if you compared your device to other existing devices and explained how your device fits into the current technology landscape. You have done this to some extent but have not used specific numbers to compare your device performance to others.
Response:
Thank you very much for your comment. This novel device has been previously described in technical detail, and compared to existing devices in other studies in the article by Du et al. (2016), while the study in this article takes the next step to explore, on one hand, the accuracy of the device in real-life clinical setting wherein the arms of HD patients on which the device is worn may make unexpected motions or movements that could cause the device to be knocked around or even dislodged, and on the other hand, the thoughts and feelings of HD patients and nurses about the using the device through questionnaires. Additional relevant contents have been added to the Discussion section on page 10.
2. Also you titled this paper "Design and Clinical Test..." but it appears you described the design of the device elsewhere and this was just the testing of the device. You may want to adjust the title so the purpose of the paper is more clear.
Response:
Thank you very much for your comment. We change the paper title to “A Clinical Trial of the Effect of a Blood Leakage Detection Device for Patients during Hemodialysis”.
Reviewer 2 Report
The reported work is interesting and important to evaluate patients' compliance, and it will be helpful to assess effectiveness of the treatment especially for those who may need hemodialysis on regular basis. However, the data analysis is based on sets of questionnaires which were designed to reflect the respondents’ subjective thoughts on the blood leakage detection device. It would be better to include blood leakage detection data collected in more objective way. Other issues:
More details are needed for Figs.1-3. Although the device has been previously reported in an earlier paper, it is worthwhile for the readers' interest to briefly introduce the working principles, the components and how the system works, what is the alarm triggering principle.
Please carefully check the figure/table caption and referred number, for example, "Table 3" should be "Table 2" in line 157. "Table 4" should be "Table 3" in line 160.
What does "10/0" mean in Table 3?
Author Response
Responses to Reviewer 2#
Paper Number | : | ijerph-538775 |
Paper Title | : | A Clinical Trial of the Effect of a Blood Leakage Detection Device for Patients during Hemodialysis |
Authors | : | Yang-Kun Ou, Ming-Jui Wu, Wei-Siang Ciou and Yi-Chun Du |
Corresponding author | : | Yi-Chun Du
|
Dear Supervisor,
We highly appreciate the reviewers’ valuable comments. The manuscript has been carefully revised accordingly. The manuscript has been revised and reformulated with good scientific rules of English to improve its quality and has been confirmed by a native English speaker. Notably, the revised and added contexts of the manuscript are marked in Red. The point-to-point responses are shown below.
For Reviewer 2#
Question 1: The reported work is interesting and important to evaluate patients' compliance, and it will be helpful to assess effectiveness of the treatment especially for those who may need hemodialysis on regular basis. However, the data analysis is based on sets of questionnaires which were designed to reflect the respondents’ subjective thoughts on the blood leakage detection device. It would be better to include blood leakage detection data collected in more objective way. Other issues:
Response:
Thank you very much for your comment. This study evaluates the device in a clinical setting on patients and explores the subjective thoughts and feelings of HD patients and nursing staff: Table 5 presents the objective findings during clinical testing, while Table 6 shows the accuracy rate of this device to be 98.9%. Findings about the subjective thoughts and feelings are shown in Tables 7 and 8 and described in page 9 lines 185-189.
Question 2: More details are needed for Figs.1-3. Although the device has been previously reported in an earlier paper, it is worthwhile for the readers' interest to briefly introduce the working principles, the components and how the system works, what is the alarm triggering principle.
Response:
Thank you for the comment. Section 2.1 Phantom arm simulation experiment has been re-written as follows on pages 2-3 lines 87-97:
“Phantom arms are one of the most commonly used teaching aids in basic clinical practices. For this experiment, a simulated blood circulation system was adopted for the phantom arm to simulate blood flow in a real arm during HD (Fig. 1). This system consisted of a single-phase 110V/60Hz adjustable-speed motor (100 mL/min−500 mL/min) (Orientalmotor, Speed Control Motor, New Taipei, Japan), and an AC110/60Hz adjustable timer (ANLY, TRD-NC relay, Taipei, Taiwan) relay with a 0.2-sec reset time (adjustable range of 0.2–5 s), enabling the system to simulate pulses of 50−180 beats per minute. In order to verify the feasibility of the simulation system, it was combined with a flow meter, a Doppler ultrasound for flowrate measurement, and heart rate variability to verify the accuracy of the simulation system. Du et al. (2016) performed tests on the system’s functions and found that the simulated blood flow velocity was similar to the default value of the dialysis machine, therefore verifying that the system could effectively simulate the conditions of HD.”
and on page 3 lines 102-106:
“In order to test for the sensitivity of the blood leakage detection system, blood was set to leak at amounts of 0.1, 0.2, 0.3 and 0.5 mL (Fig. 2) at angles of 0°, 36°, 72°, 108° , 144°, 180°, 216°, 252°, 288°, and 324° in relation to the needle – thereafter referred to as angles 1−10, respectively (Fig. 3), and the alarm was triggered when detected blood leakage in any angle.”
Question 3: Please carefully check the figure/table caption and referred number, for example, "Table 3" should be "Table 2" in line 157. "Table 4" should be "Table 3" in line 160.
Response:
Thank you for the comment. We have adjusted the former Table 4 to Table 1 in line 134, the former Table 3 to Table 2 in line 160, and the former Table 4 to Table 3 in line 163.
Question 4: What does "10/0" mean in Table 3?
Response:
The 10/0 means Success/Failure (S/F). We corrected the typo in the table 4.
Reviewer 3 Report
This study by Ou et al is an important addition to the literature on bloodleak detection devices in hemodialysis. Bloodleakage is relatively common and indeed cases with fatal ending have been described. Their study is very well written, the methods and results are clear. The addition of the reactions of both patients and nurses to the device is an important one. The discussion is brief and to-the-point.
Major points
1. The only thing I miss in the study is the cost aspect. How much does the device cost per patient? The VenAcc Patch provided by Fresenius for example costs 140 euros for 40 patches (3.5 euros per dialysis session) excluding the electrodes which can be re-used. The cost per patch is not significant when considering the lines and filters which are much more expensive.
Could the authors comment on this?
2. In the Netherlands we only use bloodleakage detection devices during nightly hemodialysis because people might move when they sleep. During the day enough staff is present to monitor the patients on bloodleakage. After a short discussion with the nursing staff in our hospital, they do not think it would be valuable to use such a device during the day when patients are awake.
Based on their study, will the authors implement their system on a standard basis?
Author Response
Responses to Reviewer 3#
Paper Number | : | ijerph-538775 |
Paper Title | : | A Clinical Trial of the Effect of a Blood Leakage Detection Device for Patients during Hemodialysis |
Authors | : | Yang-Kun Ou, Ming-Jui Wu, Wei-Siang Ciou and Yi-Chun Du |
Corresponding author | : | Yi-Chun Du
|
Dear Supervisor,
We highly appreciate the reviewers’ valuable comments. The manuscript has been carefully revised accordingly. The manuscript has been revised and reformulated with good scientific rules of English to improve its quality and has been confirmed by a native English speaker. Notably, the revised and added contexts of the manuscript are marked in Red. The point-to-point responses are shown below.
For Reviewer 3#
This study by Ou et al is an important addition to the literature on blood leak detection devices in hemodialysis. Blood leakage is relatively common and indeed cases with fatal ending have been described. Their study is very well written, the methods and results are clear. The addition of the reactions of both patients and nurses to the device is an important one. The discussion is brief and to-the-point.
Major points
Question 1: The only thing I miss in the study is the cost aspect. How much does the device cost per patient? The VenAcc Patch provided by Fresenius for example costs 140 euros for 40 patches (3.5 euros per dialysis session) excluding the electrodes which can be re-used. The cost per patch is not significant when considering the lines and filters which are much more expensive. Could the authors comment on this?
Response:
Thank you very much for your comment. The cost estimated from the bill of material (BOM) is around $0.40-0.50 USD per sensor patch in Taiwan, which is a very competitive cost, and could even be further reduced if mass-produced in the future. However, there are many other costs to consider when putting a medical device on the market, which are beyond the scope of this study.
Question 2: In the Netherlands we only use blood leakage detection devices during nightly hemodialysis because people might move when they sleep. During the day enough staff is present to monitor the patients on blood leakage. After a short discussion with the nursing staff in our hospital, they do not think it would be valuable to use such a device during the day when patients are awake. Based on their study, will the authors implement their system on a standard basis?
Response:
Thank you for sharing with us the experience in your country. In response to your question, we have added the following in the Discussion section on page 10 lines 214-222:
“In Taiwan, most HD patients undergo hemodialysis at daytime, and many fall asleep during the 3-4 hour long session, during which the occasional motions and movements of the body or limbs is a great risk for VND. Furthermore, patients at high risk for being agitated or restless – such as elderly patients with dementia – require constant attention. There was even a case of fatal hemorrhaging from VND where a long-term HD patients suffering from major depression pulled out his own needle, and reports such as these put on considerable stress on HD clinical care personnel in Taiwan. Although the HD staff-to-patient ratio is 1:4, the nursing staff may need to leave temporarily for other clinical duties and are not able to remain at the patient’s side throughout the entire HD session, therefore this device can provide significant help to improve patient safety.”
We have also added related contents in Section 3.2. Clinical test on page 8 lines 169-174:
“In this study, a total of eleven subjects with average age and standard deviation (SD) of 65.0 ± 14.2 were recruited from the Kaohsiung Veterans General Hospital (KVGH), Tainan Branch, and the clinical test design was approved by the KVGH Institutional Review Board (IRB No. VGHKS16-CT8-20). Although the clinical testing period was 5 months long and all subjects were scheduled for 3 HD sessions per week, some subjects joined the study later, therefore not everyone was tested for the full 5 months. “
Round 2
Reviewer 2 Report
The major comments have been addressed, and the revised paper can be accepted.